# High-Precision Acceleration Measurement System Based on Tunnel Magneto-Resistance Effect [note 1]

**DOI:** 10.3390/s20041117

**Published:** 2020-02-18

**Authors:** Lu Gao, Fang Chen, Yingfei Yao, Dacheng Xu

**Affiliations:** 1School of Electronic and Information Engineering, Soochow University, Suzhou 215006, China; 20175228002@stu.suda.edu.cn (L.G.); 20174228012@stu.suda.edu.cn (Y.Y.); 2Shanghai Institute of Microsystem and Information Technology, Chinese Academy of Sciences, Shanghai 200050, China; fangchen@mail.sim.ac.cn

**Keywords:** acceleration measurement, tunnel magneto-resistance effect, 1/f noise suppression

## Abstract

A high-precision acceleration measurement system based on an ultra-sensitive tunnel magneto-resistance (TMR) sensor is presented in this paper. A “force–magnetic–electric” coupling structure that converts an input acceleration into a change in magnetic field around the TMR sensor is designed. In such a structure, a micro-cantilever is integrated with a magnetic field source on its tip. Under an acceleration, the mechanical displacement of the cantilever causes a change in the spatial magnetic field sensed by the TMR sensor. The TMR sensor is constructed with a Wheatstone bridge structure to achieve an enhanced sensitivity. Meanwhile, a low-noise differential circuit is developed for the proposed system to further improve the precision of the measured acceleration. The experimental results show that the micro-system achieves a measurement resolution of 19 μg/√Hz at 1 Hz, a scale factor of 191 mV/g within a range of ± 2 g, and a bias instability of 38 μg (Allan variance). The noise sources of the proposed system are thoroughly investigated, which shows that low-frequency 1/f noise is the dominant noise source. We propose to use a high-frequency modulation technique to suppress the 1/f noise effectively. Measurement results show that the 1/f noise is suppressed about 8.6-fold at 1 Hz and the proposed system resolution can be improved to 2.2 μg/√Hz theoretically with this high-frequency modulation technique.

## 1. Introduction

High-precision measurement of acceleration, which achieves ng or μg measurement resolution, is especially important in some fields such as gravity-field tests [1], navigation systems [2,3,4], seismology detection [5], and robotics [6]. Despite substantial progress in acceleration sensing techniques, acceleration measurement to achieve navigation-level precision is still a challenge.

A variety of methods to measure acceleration with high resolution were proposed and implemented based on principles of mechanical [7], optical [8,9], and electrical detection. The quartz flexure accelerometer measures acceleration by detecting changes in the relative position of the sensitive quartz pendulum, which has low friction torque, mechanical bias error, and temperature drift. However, its structure is difficult to miniaturize [10]. The optical accelerometer measures the acceleration based on optical detection technology with advantages of strong anti-electromagnetic interference capability. However, there is a challenge for optical accelerometers to integrate with microelectronics due to complex optical modulation and demodulation units [8,9]. 

With the development of micro-electromechanical system (MEMS) technology, MEMS accelerometers based on different sensing principles [11,12] and structural designs [13,14] were developed. Capacitive-based sensing is a common method of MEMS acceleration measurement. Many optimized mechanical structural layouts [15] and interface circuit designs [16] for high-precision acceleration measurement were studied based on MEMS capacitance-based accelerometers. Colibrys (Neuchatel, Switzerland) Ltd. reported a capacitive accelerometer based on a three-stack silicon wafer structure and interfaced with a fifth-order sigma-delta control circuit that achieved a resolution of 1.7 μg/√Hz at a bandwidth of 300 Hz and long-term stability of ±100 μg [17]. This kind of closed-loop capacitive accelerometer solves the problem of nonlinear changes in capacitance and the difficulty of carrying out large dynamic range. Its measurement bandwidth can be extended through a force-rebalancing feedback loop [18]. However, the complex force-rebalancing circuit consumes additional power, which is not suitable for some low-power applications [19]. In addition, the device sensitivity is difficult to improve due to limitations in MEMS fabrication technology [20]. 

Acceleration measurement based on magnetic sensing was also exploited. Currently, magnetic sensing-based accelerometers are developed based on giant magneto-resistance (GMR). Phan from the Eindhoven University of Technology proposed a biaxial accelerometer based on GMR that achieved a sensitivity of 0.32 V/g [21]. The TDK Corporation applied GMR to measure the acceleration of head motion in discs [22]. However, GMR-based acceleration measurement has the limitations of low precision and complicated structure. The tunnel magneto-resistance (TMR) sensor, a thin film device based on magnetic tunnel junction (MTJ), exhibits extremely high sensitivity of magnetic sensing, and it was successfully applied in microelectronic memory devices [23]. Compared with anisotropic magneto-resistance (AMR) and giant magneto-resistance (GMR), the magnetic sensitivity of MTJ-based TMR exhibits a much higher (at least 10-fold) sensitivity [24,25,26]. Some further progress was achieved in improving the signal-to-noise ratio (SNR) [27] and micro-fabrication technologies [28] of TMR. Although the TMR-based sensing technique now has advantages of ultra-high sensitivity, high SNR, large dynamic range, and the capability of easy miniaturization, research of acceleration measurement based on TMR sensing is still seldom reported. 

This paper expands on preliminary research presented in Reference [29]. The detailed analysis and implementations of the micromechanical structure and acceleration measurement system are further presented. In particular, an advanced noise suppression method is proposed, which takes the TMR-based acceleration measurement system close to an ultra-low 1/f noise floor. The key design of a TMR acceleration measurement system converts an acceleration input to a change in magnetic field around the TMR sensor. Here, we design and fabricate a “force–magnetic–electric” coupling structure combined with a highly sensitive TMR sensor. In such a structure, a micro-cantilever is integrated with a magnetic field source on its tip. Under an acceleration, the mechanical displacement of the cantilever causes a change in the spatial magnetic field sensed by the TMR sensor. With the proposed structure, a large linear dynamic detection range is achieved. Furthermore, the TMR sensor used in the proposed system is designed as a Wheatstone bridge structure consisting of four tunnel magneto-resistances, which have the advantage of high sensitivity. The noise sources, especially related to the low-frequency noise, of the proposed system are thoroughly investigated for system optimization. A strategy to suppress the 1/f noise of the TMR sensor is proposed and analyzed to further improve the measurement resolution. A high precision and low noise acceleration measurement were achieved by the proposed system.

The remainder of this paper is organized as follows: Section 2 describes the overall system design of the acceleration measurement system based on the TMR effect, importantly focusing on the introduction of the “force–magnetic–electric” coupling principle. A detailed analysis of the force field coupling structure and magnetic field distribution is carried out, and the circuit design for signal processing is introduced. Section 3 focuses on the test process and experimental results of the micro-acceleration measurement system. Discrepancies between theoretical design and the results of a practical test are analyzed in detail. Section 4 describes noise sources of the micro-system, and the implementation of low-frequency 1/f noise suppression to improve the measurement resolution of the micro-system is explored. Section 5 provides the conclusions of this paper.

## 2. Design of the Acceleration Measurement System

The structure of the proposed micro-system is shown in Figure 1. It consists of a micro-cantilever, a permanent magnet, a TMR sensor, and a signal processing circuit. The input acceleration *a* causes the micro-cantilever and the magnet attached to its tip to produce a displacement Δ*Z* along the *Z*-direction. As a result, the strength of the distributed magnetic field along the *X*-direction varies by Δ*H* with the displacement of the magnet. The change in the magnetic field is then converted into the magneto-resistance value change Δ*R*, through a TMR sensor constructed in a bridge structure. Interfaced with a low-noise signal processing circuit, the change in the magneto-resistance value is transformed into a voltage signal Δ*V* to realize acceleration measurement. The analysis of the “force–magnetic–electric” coupling structure is described below. 

### 2.1. Design of Force Field Coupling Structure

The coupling structure of the force field is based on the design of the micro-cantilever structure to realize the conversion from input acceleration to mechanical displacement [30,31].

The load distribution model of the cantilever with a mass is shown in Figure 2. In this model, the self-weight of the cantilever is assumed to uniformly distribute along the length, and the magnetic field source is equivalent to a uniform load applied within the area where the magnet is attached. The deflection [32] at the tip of the cantilever under an input acceleration (*a_input_*) can be obtained as
(1)$$\Delta Z=\left(-\frac{3m_{beam}l^{3}}{2bh^{3}E}-\frac{4m_{magnet}l^{3}}{bh^{3}E}+\frac{3m_{magnet}wl^{2}}{bh^{3}E}\right)\cdot a_{input},$$
where *m_beam_* and *m_magnet_* are weights of the micro-cantilever and the permanent magnet, respectively, *l*, *b*, and *h* are the length, width, and height of the micro-cantilever, respectively, and *E* is Young’s modulus.

Mechanical–thermal noise [33] limits the resolution of the micro-cantilever. The equivalent acceleration resulting from mechanical–thermal noise can be expressed as
(2)$$noise=\frac{\sqrt{4k_{B}cT}}{9.8M}\left(g/\sqrt{Hz}\right),$$
where *k_B_* is the Boltzmann constant, *T* is the absolute temperature, *c* is the damping, and *M* is the effective mass of the system. Squeeze film damping generally leads to the mechanical–thermal noise.

The dimension of the silicon micro-cantilever is 6000 × 1500 × 15 μm^3^ in this prototype micro-system. A permanent magnet made of samarium cobalt with a size of 1500 × 1500 × 500 μm^3^ is attached to the tip of the micro-cantilever. Based on the theoretical analysis, the sensitivity of the mechanical displacement due to the input acceleration (*S_cantilever_*) is 76.7 μm/g, consistent with the results of a numerical simulation of 75.7 μm/g shown in Figure 3. The mechanical–thermal noise of the designed force field coupling structure is 4.85 ng/√Hz when squeeze film damping is considered. Based on the Rayleigh–Ritz method [34,35], a self-resonant frequency of 62.37 Hz is obtained.

### 2.2. Design of Magnetic Field Source

The permanent magnet attached to the tip of the cantilever implements the conversion of a mechanical displacement into a change in the magnetic field. The equivalent magnetic charge model [36,37] shown in Figure 4a is applied to analyze the magnetic field distribution. Taking the center of the permanent magnet as the origin of the Cartesian coordinate system, the equivalent magnetic charge model regards the magnet as a surface magnetic charge with density *P_sm_* when uniformly magnetized along the *Z*-direction.

The strength of the magnetic field along the *X*-direction generated by the cube permanent magnet at the observation point *P* can be calculated as
(3)$$\begin{array}{ll} H_{x}= & \frac{1}{4\pi \mu_{0}}(\int_{-\frac{a}{2}}^{\frac{a}{2}}\int_{-\frac{b}{2}}^{\frac{b}{2}}Br\frac{\left(u-x^{\prime }\right)i}{{\left[{\left(u-x^{\prime }\right)}^{2}+{\left(v-y^{\prime }\right)}^{2}+{\left(w-h/2\right)}^{2}\right]}^{3/2}}dx^{\prime }dy^{\prime }\\ & -\int_{-\frac{a}{2}}^{\frac{a}{2}}\int_{-\frac{b}{2}}^{\frac{b}{2}}Br\frac{\left(u-x^{\prime }\right)i}{{\left[{\left(u-x^{\prime }\right)}^{2}+{\left(v-y^{\prime }\right)}^{2}+{\left(w+\frac{h}{2}\right)}^{2}\right]}^{\frac{3}{2}}}dx^{\prime }dy^{\prime }), \end{array}$$
where (*x’*, *y’*, ±*h/2*) are coordinates of the surface magnetic charge, and (*u*, *v*, *w*) are coordinates of the observation point *P*. The magnetic field distribution along the *X*-direction with a remanence of 1.023 T is shown in Figure 4b, where a linear region is marked with the dash box.

The sensitive point of the TMR sensor used in this paper is set at (1025, 0, 0) under micron units. The theoretical result and simulation of the magnetic field distribution at the sensitive point are shown in Figure 5. Therefore, the sensitivity of *Hx* due to the displacement of the magnet along the *Z*-direction (*S_magnet_*) is 5.18 Oe/μm, and that in the linear region along the *Z*-direction (△*Z_linear_*) is ±155.5 μm at the sensitive point.

### 2.3. Design of Signal Processing Circuit

The TMR sensor (TMR2105 series from MultiDimension Technology Inc.) was constructed with a Wheatstone bridge structure with a sensitivity (*S_TMR_*) of 0.93 mV/Oe. The signal processing circuit is shown in Figure 6. The TMR sensor is driven by a voltage reference with an ultra-low noise to reduce the differential-mode noise coupled into the output signal when arms of the bridge are unbalanced due to the input magnetic signal. The differential output signal of the TMR sensor is interfaced with buffers and zero-regulators to compensate for the initial imbalance of arms of the bridge. The differential output signal is then converted into the analog signal output of acceleration by a subtractor and a low-pass filter with a bandwidth of 100 Hz, which is higher than the frequency of the acceleration input. Low-noise devices and differential structure circuits are applied to the signal processing circuit to ensure a low-noise performance. The circuit noise model [38] is shown in Figure 6, where vn*x*(x = 1, …, 8) is the input voltage noise and In*x*(x = 1, …, 16) is the input current noise. Based on the combination of input voltage noise, input current noise, and thermal noise, the output noise is theoretically calculated as 22.23 nV/√Hz.

In summary, the system’s sensitivity (*S_system_*) and range of measurement (*D_system_*) can be expressed as follows:(4)$$S_{system}=S_{cantilever}\cdot S_{magnet}\cdot S_{TMR},$$
(5)$$D_{system}=\frac{\Delta Z_{linear}}{S_{cantilever}}.$$

The sensitivity of the entire system is 370 mV/g with a measurement range of ±2 g based on the “force–magnetic–electric” coupling structure.

## 3. Results and Discussion

The micro-cantilevers were fabricated with an in-house bulk micromachining process. Figure 7 depicts the simplified fabrication process using a double-side polished n-type (100) four-inch silicon wafers, with a resistivity of 1–10 Ω∙cm and a thickness of 450 μm. The process mainly involved the following steps:

(I) The SiN layers were formed at its double sides, and then patterned for cantilevers. With low-pressure chemical vapor deposition (LPCVD), two 0.3-μm-thick SiN layers were sequentially deposited on both sides of the silicon wafer. 

(II) After photolithography, the micro-cantilever features were transferred to the (100) silicon wafer by deep reactive ion etching (DRIE). Both sides of the wafer were patterned and etched by DRIE: the frontside with trenches to form device features and gaps of the acceleration measurement system; the backside that allowed removing the handle wafer block. The backside wafer was patterned with a 10-μm-thick AZ9260-type positive photoresist, followed by DRIE down to 200 μm to define the depth on the handle part. 

(III) After the backside etching, the frontside photoresist was patterned. The frontside part was patterned by a diluted 3-μm AZ9260 photoresist and etched in DRIE down to 15 μm to define the cantilever thickness and the gap between the cantilever and TMR sensor. Here, DRIE was used again to further vertically etch both sides of the wafer to define the thickness and gap of the structure. 

The micro-system was in a 14 × 11 × 2 mm^3^ package, and a photograph of the system is shown in Figure 8. This micro-system was interfaced with a signal processing circuit, as shown in Figure 6.

The micro-system was fixed on the base of the high-precision centrifuge LXJ-900. Its direction of measurement was toward the rotating shaft along the radius of the turntable. The output of the analog signal was connected to a 24-bit data acquisition card NI 9219 through a shielded cable controlled using NI Labview software. The results of the dynamic response test are shown in Figure 9. The system showed a measurement range of ±2 g, with a sensitivity of 181 mV/g. Because of a signal attenuation of −0.46 dB after compensating for the initial imbalance in arms of the bridge, the system’s sensitivity was 191 mV/g, lower than the theoretical value of 370 mV/g as discussed in Section 2.3.

The assembly of the micro-system was observed by an electron microscope as shown in Figure 10. There was an error during the assembly process. Taking the center of the magnet as the origin of the Cartesian coordinate system, the coordinates of the equivalent sensitive point of the TMR sensor were (1169, −198.5, 0), far from the designed position (1025, 0, 0) under micron units. Considering the actual sensitive position, the sensitivity of *Hx* due to the displacement of the magnet along the *Z*-direction was 2.72 Oe/μm and that in the linear region along the *Z*-direction was ±161 μm based on Equation (3). Finally, the system’s sensitivity was 194 mV/g and its range of measurement was ±2 g when considering assembly error, which is consistent with the experimental results.

The resolution of acceleration measurement is limited by system noise. The sources of noise in the system were investigated. The noise experiment was based on a dynamic signal analyzer Agilent 35670A. The test results of system noise are shown in Figure 11. The noise floor of the signal processing circuit was 40 nV/√Hz, which is close to the theoretical output noise and corresponded to an acceleration resolution of 0.2 μg/√Hz. The noise floor of the signal processing circuit had little impact on the sensing resolution. When the arms of the bridge of the TMR sensor were unbalanced due to the change in the magnetic field caused by the input acceleration signal, the bias voltage noise was converted from a common-mode noise to a differential-mode noise, which could be coupled with the output signal. The noise of the bias voltage of the TMR sensor was 65 nV/√Hz, corresponding to an acceleration resolution of 0.36 μg/√Hz. Then, a part of the noise of the bias voltage coupled to the output signal had negligible impact on the sensing resolution. It is clear from the figure that the output noise of the system exhibited a significant 1/f trend. The system’s noise was about 3.45 μV/√Hz at 1 Hz, corresponding to an acceleration resolution of 19 μg/√Hz at 1 Hz. These results show that the main noise source of the system in the low-frequency region was the 1/f noise of the TMR sensor, which limited the resolution of acceleration measurement. There was an obvious peak at 62 Hz originating from the resonant frequency of the force field coupling structure that limited the measurement bandwidth of the system. The noise spiked at 50 Hz and its multiple harmonics resulted from power line interference. Moreover, by integrating the system noise shown in Figure 11 with the bandwidth of 100 Hz set by the low-pass filter, the peak-to-peak noise of the system was 76.4 μV, corresponding to an acceleration resolution of 422 μg.

As shown in Figure 12, the micro-system was fixed on the base of a high-precision three-dimensional turntable 3st-600. The resolution of the turntable was one arc second, corresponding to a gravity projection of 4.84 μg. The angle of rotation of the turntable was changed through a program control software for an input acceleration step of 500 μg. The results of the resolution test are shown in Figure 13, and the empirical resolution was approximately 500 μg, comparable to the theoretical resolution calculated based on the peak-to-peak noise. Reducing the bandwidth of the signal processing circuit could further improve the system resolution. As shown in Figure 13, there was prominent magnetic hysteresis in the system response test, which caused a response discrepancy of the TMR sensor under the same input acceleration. Therefore, suppressing the magnetic hysteresis of the TMR sensor to improve the performance of system response will be a focus of future work.

Bias stability is defined as the system’s output (1σ value) measured over a specified period under zero input acceleration. The acceleration measurement system was first warmed at room temperature for 1 h. Figure 14 shows the measured acceleration output for 1 h at a sampling frequency of 100 Hz. The bias stability was calculated as 266 μg by taking a 1σ deviation. The temperature variation and low-frequency noise were the main factors restricting the bias stability of the acceleration system. This experiment was performed at room temperature, and no temperature compensation algorithm was used. Therefore, the temperature characteristics of the TMR sensor should be explored to improve bias stability. The bias instability represents the lowest bias error that the accelerometer can achieve, which is defined as the floor of the Allan variance curve. The Allan variance calculated from the analog output is shown in Figure 15, and the bias instability was 38 μg. Bias instability was mainly determined by the 1/f noise of the micro-system. The suppression of the 1/f noise of the TMR sensor is critical for improving sensing resolution and bias instability. 

Bias stability is defined as the system’s output (1σ value) measured over a specified period under zero input acceleration. The acceleration measurement system was first warmed at room temperature for 1 h. Figure 14 shows the measured acceleration output for 1 h at a sampling frequency of 100 Hz. The bias stability was calculated as 266 μg by taking a 1σ deviation. The temperature variation and low-frequency noise were the main factors restricting the bias stability of the acceleration system. This experiment was performed at room temperature, and no temperature compensation algorithm was used. Therefore, the temperature characteristics of the TMR sensor should be explored to improve bias stability. The bias instability represents the lowest bias error that the accelerometer can achieve, which is defined as the floor of the Allan variance curve. The Allan variance calculated from the analog output is shown in Figure 15, and the bias instability was 38 μg. Bias instability was mainly determined by the 1/f noise of the micro-system. The suppression of the 1/f noise of the TMR sensor is critical for improving sensing resolution and bias instability. 

The structure parameters of the designed acceleration measurement system based on the TMR sensor are shown in Table 1 and the measurement results are summarized in Table 2. Compared to the *Z*-axis tunneling magneto-resistive accelerometer fabricated by Southeast University with a sensitivity of 8.85 mV/g and a noise floor of 86.2 μg /√Hz [39], the proposed system achieved a higher sensitivity and better noise performance. With the low-frequency 1/f noise suppression method described in Section 4, the measurement resolution could be further improved.

## 4. System Noise Suppression

### 4.1. Analysis of System Noise

The overall resolution of the micro-system is determined by the mechanical–thermal noise, interface circuit noise, TMR noise, and magnetic noise. The calculated mechanical–thermal noise of the designed force field coupling structure was 4.85 ng/√Hz. The measured interface circuit noise and bias voltage noise were 40 nV/√Hz and 65 nV/√Hz, equivalent to an acceleration resolution of 220 ng√Hz and 360 ng/√Hz, respectively. Magnetic noise mainly originates from the non-uniformity of the magnetic field distribution and geomagnetic interference [40]. However, the spatial distribution of the static magnetic field generated by the permanent magnet had a highly uniform gradient, and the package with high magnetic permeability could effectively suppress geomagnetic interference. Therefore, the mechanical–thermal noise of the force field coupling structure, the magnetic noise of the magnetic field distribution, and the electrical noise of the bias voltage and interface circuit were not the main factors affecting the resolution of the system. The sensing resolution of this micro-system was mainly limited by the low-frequency noise of the TMR sensor. This was the focus while exploring methods of 1/f noise suppression.

### 4.2. Modulation of Driving Source of TMR

The high-frequency modulation technique of the driving source of the TMR sensor is shown in Figure 16. The basic principle is that the TMR sensor was driven by a high-frequency voltage source so that the output of the TMR sensor was modulated to the frequency of the driving source. Then, the operating point of the TMR sensor could be modulated from the low-frequency region dominated by the 1/f noise to the high-frequency region dominated by thermal noise to improve the resolution of sensing. The analog signal at the operating point in the high-frequency region could be mixed by the clock of the driving source and low-pass filtered to obtain the desired analog acceleration information.

The experiment was based on a millimeter-scale TMR sensor-based acceleration measurement device with a stainless-steel skeleton [41]. The magnetic field was generated from the permanent magnet. The low-frequency noise of the system’s output with the TMR sensor biased by a direct current (DC) of 3 V is shown in Figure 17a, compared with the effect of 1/f noise suppression of the modulation technique. The high-frequency driving source was set to 2 V + 0.5 V sin(2π3000t). The amplitude of the driving source was in the region of operation of the TMR sensor, and its modulation frequency was higher than the corner frequency [42,43] of the TMR sensor 1/f noise. 

The low-frequency noise (0–200 Hz) and high-frequency noise (2900–3100 Hz) measured from the output of the TMR sensor after modulation are shown in Figure 17b,c, respectively. There was significant 1/f noise in the low-frequency region (0–200 Hz) originating from the DC component of the driving source. The peak at 3 kHz in Figure 17c is a modulated signal, indicating that the operating point of the TMR sensor was modulated to the high-frequency region through driving source modulation. The modulated signal was mixed by the same clock of the driving source and low-pass filtered to obtain the desired signal with noise shown in Figure 17d. However, the noise of the demodulated signal was nearly identical to that shown in Figure 17a.

The operating point of this TMR sensor was modulated from the low-frequency region to the high-frequency region using driving source modulation. According to Figure 17c, there was still significant 1/f noise around the modulated signal at 3 KHz. Although the operating point of the TMR sensor was modulated, 1/f noise was also modulated from the low-frequency to the high-frequency region after driving source modulation. Figure 17d indicates that, after demodulation and low-pass filtering, the measured signal demodulated by the mixer still had 1/f noise.

The process of 1/f noise suppression using driving source modulation based on this micro-system can be described as follows:

Suppose the TMR sensor is biased by DC voltage. The signal output due to the magnetic field from the input acceleration is
(6)$$B\left(t\right)=a_{input}S_{cantilever}S_{magnet}S_{TMR}\sin\left(w_{0}t\right)+A_{noise}\left(w_{m}t\right),$$
where *A_noise_* is the amplitude of the accompanying noise, *w_0_* is the frequency of the low-frequency acceleration signal, and *w_m_* is the frequency of low-frequency noise.

The signal output of the TMR sensor after driving source modulation can be expressed as
(7)$$\begin{array}{ll} B_{mod}\left(t\right)= & \left[a_{input}S_{cantilever}S_{magnet}S_{TMR}\sin\left(w_{0}t\right)+A_{noise}\left(w_{m}t\right)\right]\cdot \sin\left(w_{s}t\right)\\ & =\frac{a_{input}S_{cantilever}S_{magnet}S_{TMR}}{2}\left[\cos\left(w_{0}-w_{s}\right)t-\cos\left(w_{0}+w_{s}\right)t\right]\\ & +\frac{A_{noise}}{2}\left[\cos\left(w_{m}-w_{s}\right)t-\cos\left(w_{m}+w_{s}\right)t\right], \end{array}$$
where *w_s_* is the modulation frequency.

The output of the analog signal after demodulation and low-pass filtering is
(8)$$\begin{array}{ll} LPF\left[B_{demod}\left(t\right)\right] & \\ & =LPF\left[\left(a_{input}S_{cantilever}S_{magnet}S_{TMR}\sin\left(w_{0}t\right)+A_{noise}\left(w_{m}t\right)\right)\cdot \sin\left(w_{s}t\right)\cdot \sin\left(w_{s}t\right)\right]\\ & =LPF[\frac{1}{2}\left(a_{input}S_{cantilever}S_{magnet}S_{TMR}\sin\left(w_{0}t\right)+A_{noise}\left(w_{m}t\right)\right)\\ & -\frac{a_{input}S_{cantilever}S_{magnet}S_{TMR}}{4}\left(\sin\left(w_{0}+2w_{s}\right)t+\sin\left(w_{0}-2w_{s}\right)t\right)\\ & -\frac{A_{noise}}{4}\left(\sin\left(w_{m}+2w_{s}\right)t+\sin\left(w_{m}-2w_{s}\right)t\right)]\\ & =\frac{1}{2}\left(a_{input}S_{cantilever}S_{magnet}S_{TMR}\sin\left(w_{0}t\right)+A_{noise}\left(w_{m}t\right)\right). \end{array}$$

When the TMR sensor is modulated using the driving source modulation technique, the measured signal and low-frequency noise are modulated to the high-frequency region and demodulated back to the low-frequency region at the same time. Therefore, the 1/f noise of this micro-system cannot be separated from the measured signal and effectively suppressed through electrical modulation of the driving source voltage, thus limiting the resolution of the acceleration measurement system. 

### 4.3. Modulation of Sensitive Source of TMR

The measured acceleration signal had low frequency, and it was converted into a low-frequency magnetic field signal using the permanent magnet. The magnetic signal sensed by the TMR sensor was significantly limited by 1/f noise in the low-frequency region. Thus, the high-frequency modulation of the sensitive source is proposed in Figure 18. The basic principle is that the magnetic source is converted from the permanent magnet to the coil using alternating current (AC). When the TMR sensor is driven by DC voltage, the change in the magnetic field caused by the low-frequency acceleration input is converted into a high-frequency magnetic signal. Thus, low-frequency acceleration is modulated to the same frequency of the AC current flowing inside the coil without complicated mechanical modulation. The operating point of the TMR sensor can then be modulated to the high-frequency region, where thermal noise is much smaller than the 1/f noise in the low-frequency region, which improves the precision of acceleration measurement. The modulated signal is then demodulated by the mixer and low-pass filtered to obtain the required acceleration information.

The process of 1/f noise suppression through the sensitive source modulation technique can be described as follows:

The signal output of the TMR sensor sensing the modulated signal can be expressed as follows:(9)$$B_{mod}\left(t\right)=a_{input}\sin\left(w_{0}t\right)S_{cantilever}S_{magnet}\sin\left(w_{s}t\right)S_{TMR}+A_{noise}\left(w_{m}t\right),$$
where *S_magnet_*sin*(w_s_t)* is the magnetic sensitivity of the coil with AC current, *w_0_* is the frequency of the low-frequency acceleration signal, *w_s_* is the frequency of AC current, and *w_m_* is the frequency of low-frequency noise.

The output of the analog signal after demodulation and low-pass filter is
(10)$$\begin{array}{ll} LPF\left[B_{demod}\left(t\right)\right] & \\ & =LPF\left[(a_{input}\sin\left(w_{0}t\right)S_{cantilever}S_{magnet}\sin\left(w_{s}t\right)S_{TMR}+A_{noise}\left(w_{m}t\right))\cdot \sin\left(w_{s}t\right)\right]\\ & =\mathrm{LPF}[\frac{a_{input}\sin\left(w_{0}t\right)S_{cantilever}S_{magnet}S_{TMR}}{2}\left(1-\cos\left(2w_{s}t\right)\right)\\ & +\frac{A_{noise}}{2}\left(\cos\left(w_{m}-w_{s}\right)t-\cos\left(w_{m}+w_{s}\right)t\right)]\\ & =\frac{a_{input}\sin\left(w_{0}t\right)S_{cantilever}S_{magnet}S_{TMR}}{2}. \end{array}$$

As shown in Equation (10), 1/f noise is separated from the measured signal and filtered based on the sensitive source modulation technique in theoretical analysis.

The experiment set-up was performed on a millimeter-scale TMR sensor-based acceleration measurement device with a stainless-steel skeleton. The magnetic field was generated from a coil with AC stimulation current at 3 kHz. The low-frequency noise (0–200 Hz) and high-frequency noise (2900–3100 Hz) measured from the system output are shown in Figure 19a,b, respectively. There was significant 1/f noise in the low-frequency region (0–200 Hz) originating from the DC supply voltage. The peak at 3 kHz shown in Figure 19b is the modulated signal. 

The operating point of the TMR sensor was modulated to a high frequency, and noise around the modulated signal was only 200 nV/√Hz. The modulated signal was then demodulated, and noise in the demodulated signal is shown in Figure 19c, which was much smaller than the 1/f noise shown in Figure 19a. To compensate for the signal attenuation of 2/π caused by the demodulation of the mixer, the 1/f noise of the demodulated signal was suppressed about 8.6-fold at 1 Hz through sensitive source modulation. At the same time, 1/f noise in the low-frequency region was modulated to the high-frequency region as shown in Figure 19d, where it could be readily filtered. As a result, the precision of measurement could be theoretically improved to 2.2 μg/√Hz at 1 Hz when the input of the low-frequency acceleration was transformed into the high-frequency magnetic field. Therefore, engineering the micro-system integrated with the coil driven by an AC stimulation current will also be the focus of future work.

## 5. Conclusions

In this paper, a high-precision tunnel magneto-resistance acceleration measurement system was proposed and demonstrated. The micro-system consisted of a micro-cantilever, a permanent magnet, a TMR sensor, and a signal processing circuit. By converting the input acceleration into a change of magnetic field around the TMR sensor, high-precision acceleration measurement was achieved due to the high sensitivity of the TMR sensor. The designed acceleration measurement system achieved a scale factor of 191 mV/g within a measurement range of ±2 g, a bias instability of 38 μg, and a bias stability of 266 μg. The dominant noise source of this system was the 1/f noise of the tunnel magneto-resistance sensor, which determined the resolution of acceleration measurement. Therefore, we explored an effective method of 1/f noise suppression to provide a strategy for the design of a high-precision acceleration measurement system based on the tunnel magneto-resistance effect. The results of tests show that system noise corresponded to an acceleration resolution of 19 μg/√Hz at 1 Hz. The 1/f noise of the system’s output could be suppressed about 8.6-fold at 1 Hz through sensitive source modulation. Then, the resolution of the acceleration measurement system based on the tunnel magneto-resistance effect could be theoretically improved to 2.2 μg /√Hz at 1 Hz. Therefore, designing a system that uses an AC coil as the magnetic source and improving the SNR of the magnetic field generated by the AC coil will be our focus in future research.

## Figures and Tables

**Figure 1 sensors-20-01117-f001:**
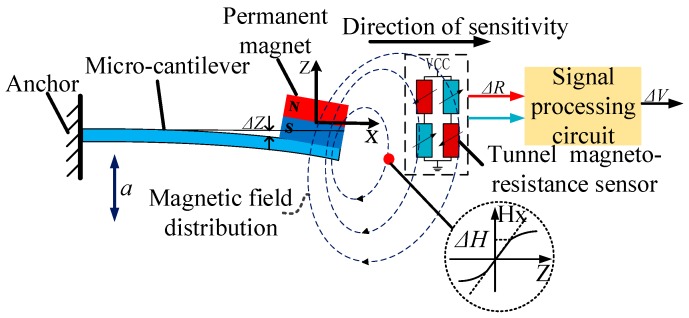
The structure of the acceleration measurement system.

**Figure 2 sensors-20-01117-f002:**
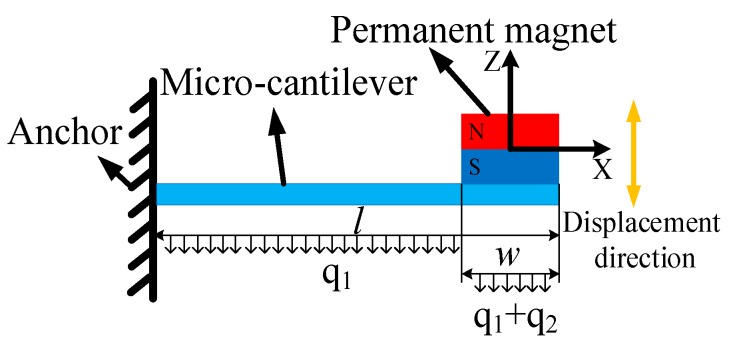
Load distribution model of force field coupling structure.

**Figure 3 sensors-20-01117-f003:**
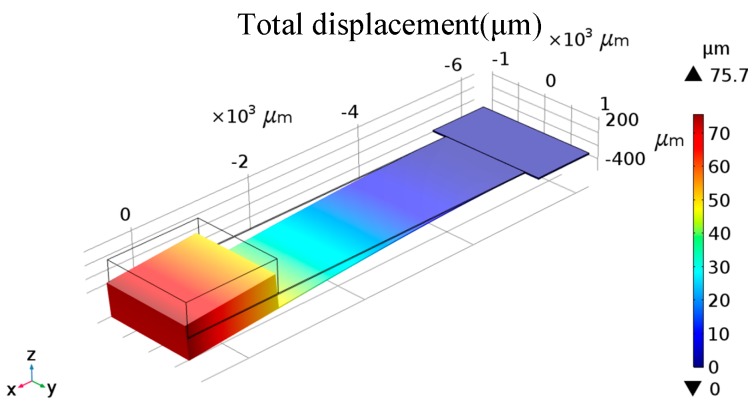
Analysis and simulation of force field coupling structure.

**Figure 4 sensors-20-01117-f004:**
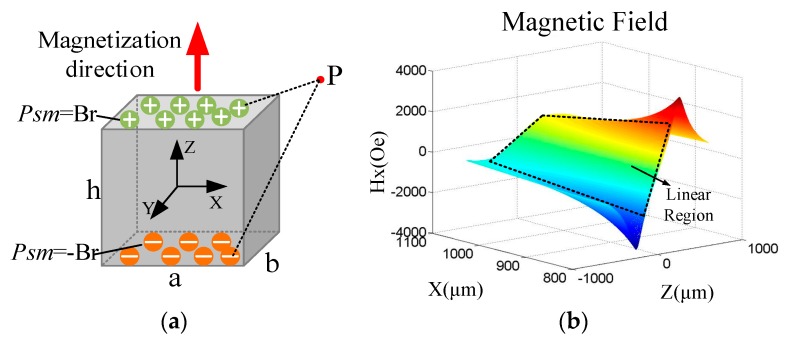
Analysis and calculation of spatial magnetic field distribution: (**a**) the equivalent magnetic charge model of the cube permanent magnet; (**b**) the magnetic field distribution along the *X*-direction of the magnet with a remanence of 1.023 T.

**Figure 5 sensors-20-01117-f005:**
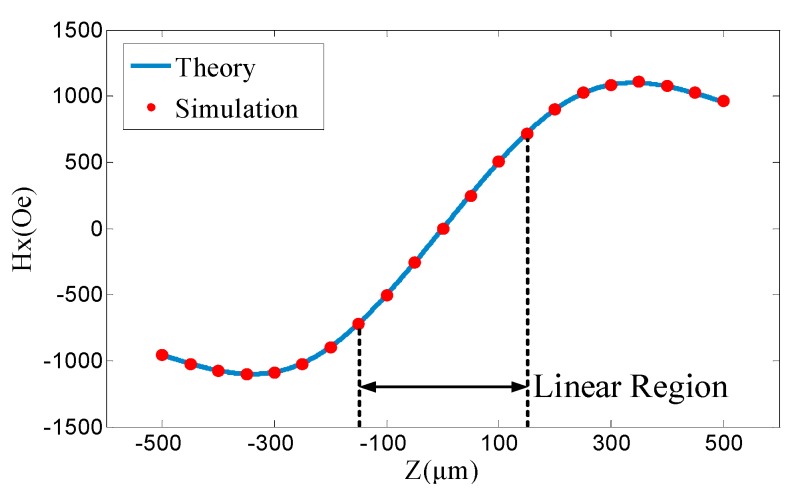
Magnetic field strength along the *X*-direction at the sensitive point.

**Figure 6 sensors-20-01117-f006:**
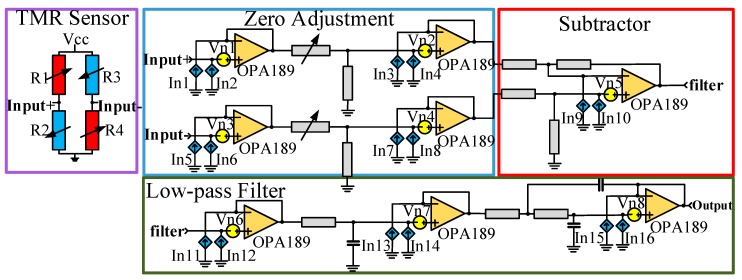
Signal processing circuit and circuit noise model.

**Figure 7 sensors-20-01117-f007:**
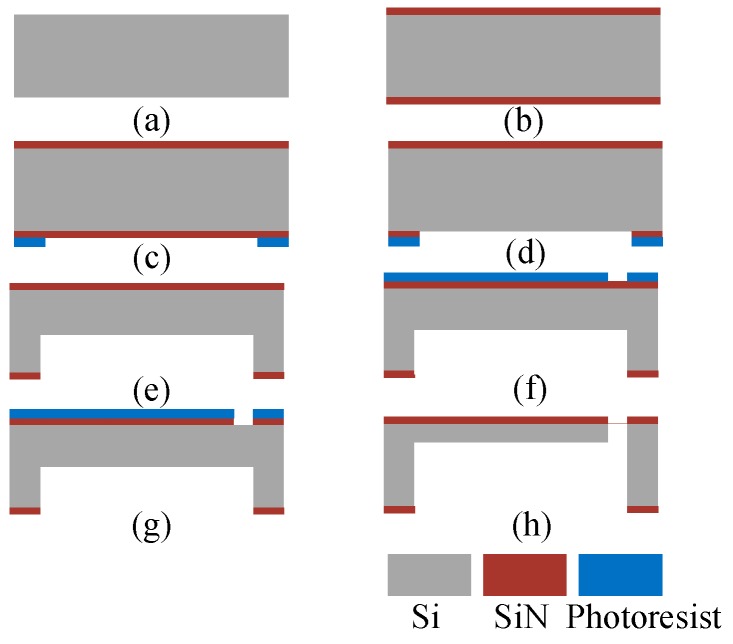
Processing flow of the silicon micro-cantilever structure: (**a**,**b**) depositing SiN layers with low-pressure chemical vapor deposition; (**c**–**e**) backside etching to form the micro-cantilever features; (**f**–**h**) etching both sides to define the thickness and the gap of the structure.

**Figure 8 sensors-20-01117-f008:**
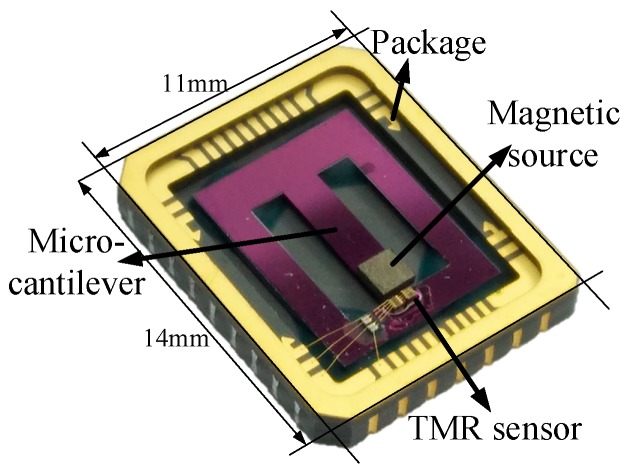
Photograph of the packaged micro acceleration measurement system.

**Figure 9 sensors-20-01117-f009:**
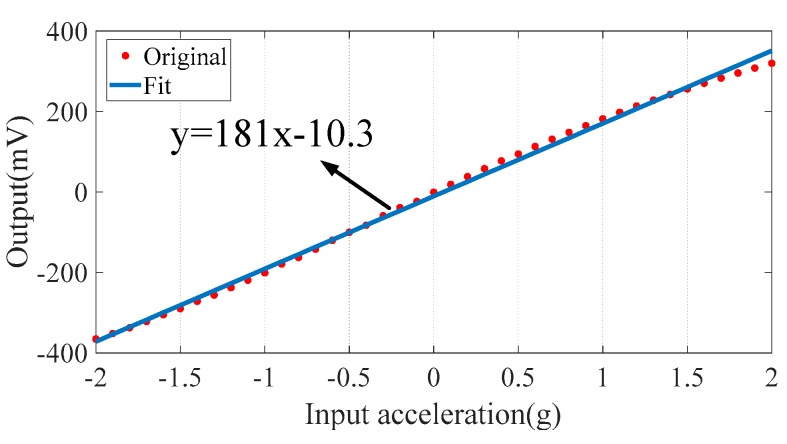
Measured response of the micro-system.

**Figure 10 sensors-20-01117-f010:**
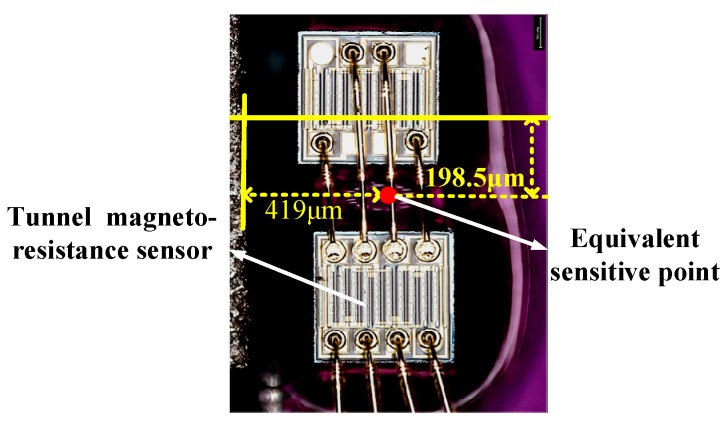
Photograph of assembly of the micro-system.

**Figure 11 sensors-20-01117-f011:**
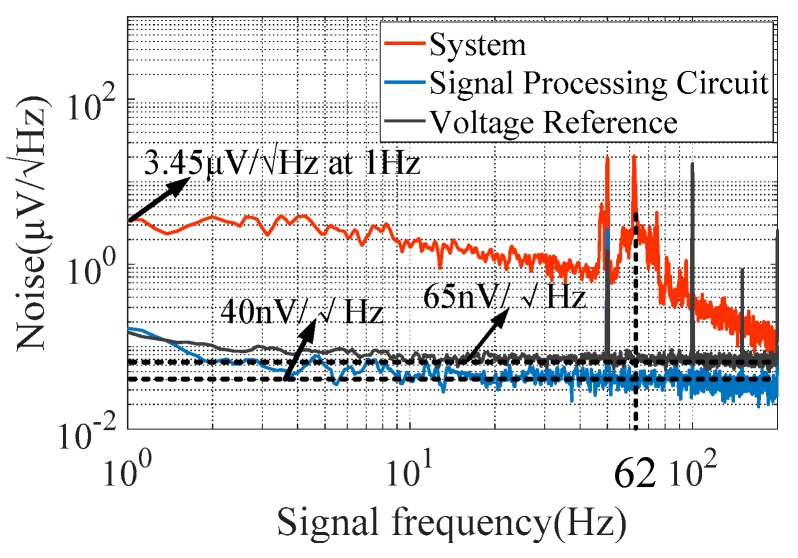
Noise characteristics of the micro-system.

**Figure 12 sensors-20-01117-f012:**
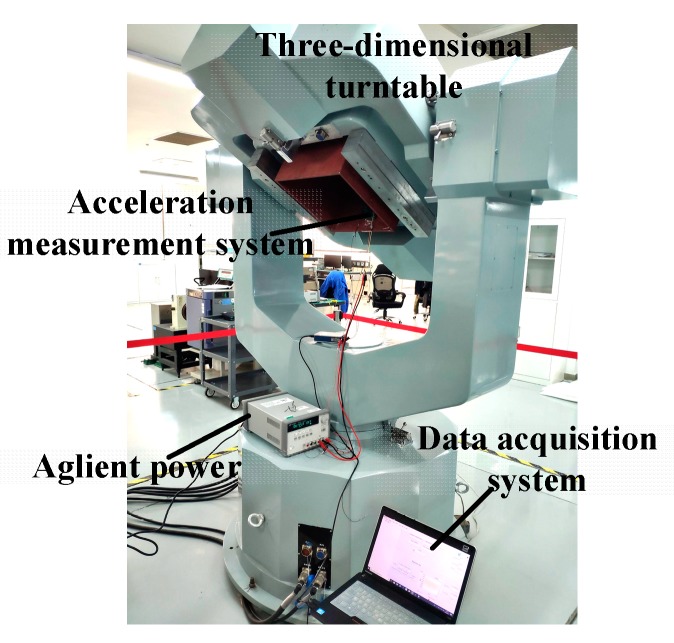
Tests on high-precision three-dimensional turntable.

**Figure 13 sensors-20-01117-f013:**
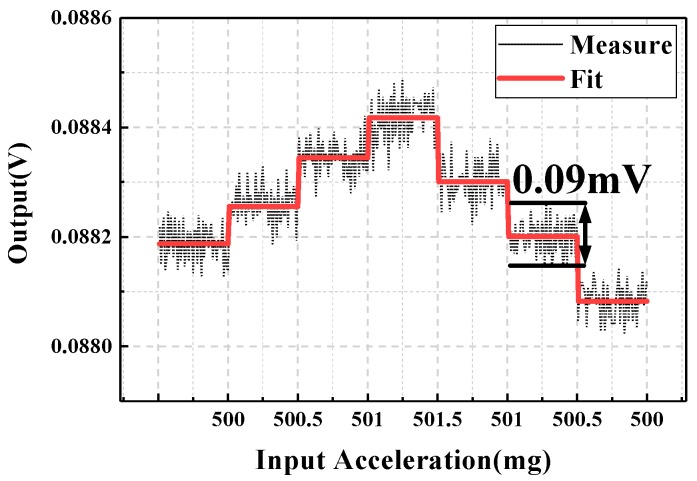
Result of resolution test.

**Figure 14 sensors-20-01117-f014:**
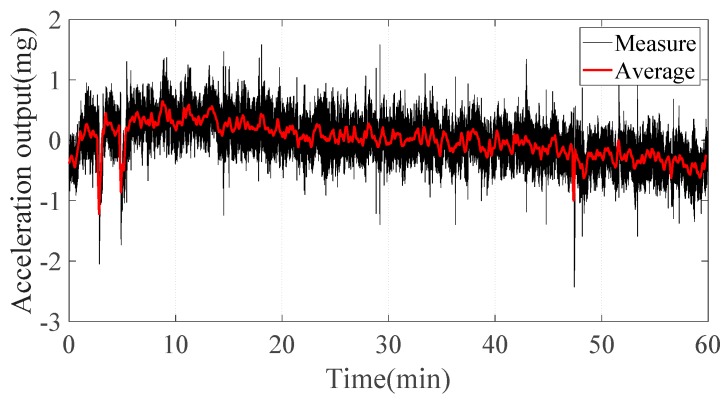
Zero-acceleration output at room temperature.

**Figure 15 sensors-20-01117-f015:**
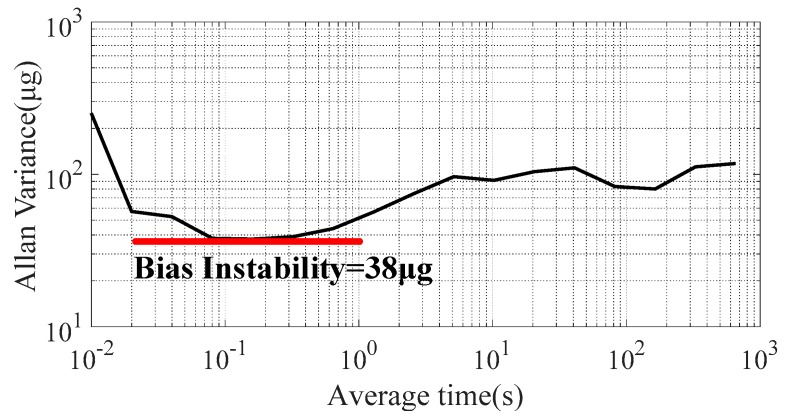
Measured Allan variance plot.

**Figure 16 sensors-20-01117-f016:**
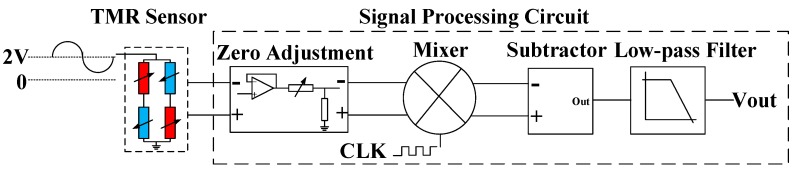
Driving source modulation.

**Figure 17 sensors-20-01117-f017:**
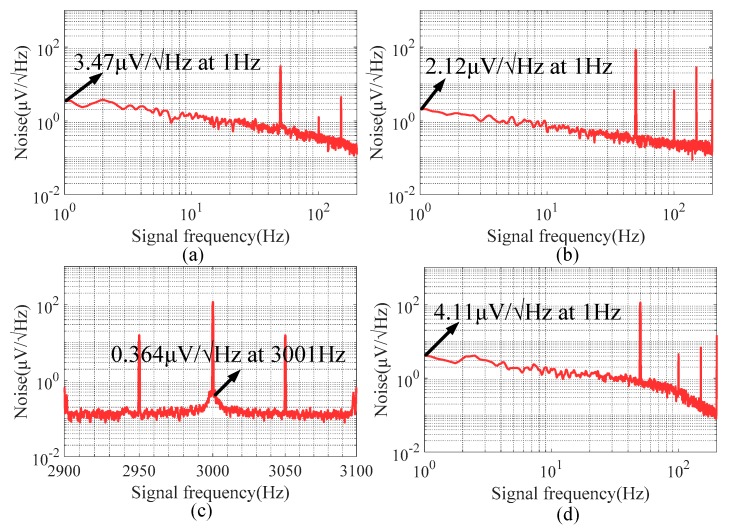
(**a**) Noise of TMR biased by direct current (DC) voltage. (**b**) Noise of TMR biased by alternating current (AC) voltage (0–200 Hz). (**c**) Noise of TMR biased by AC voltage (2900–3100 Hz). (**d**) Noise of TMR after demodulation.

**Figure 18 sensors-20-01117-f018:**
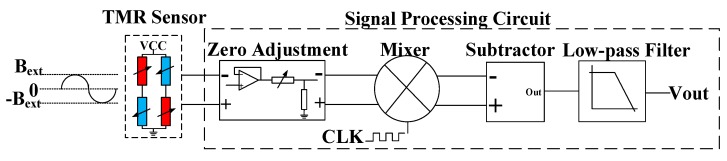
Sensitive source modulation.

**Figure 19 sensors-20-01117-f019:**
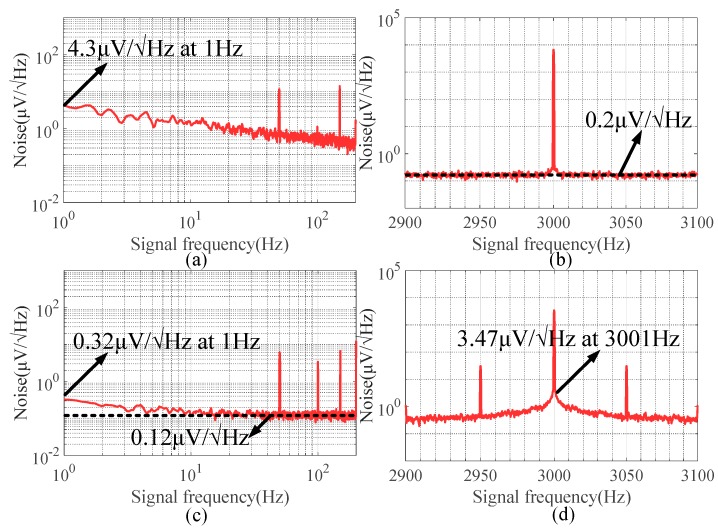
(**a**) Noise of TMR exposed to AC magnetic field (0–200 Hz). (**b**) Noise of TMR exposed to AC magnetic field (2900–3100 Hz). (**c**) Noise of TMR after demodulation (0–200 Hz). (**d**) Noise of TMR after demodulation (2900–3100 Hz).

**Table 1 sensors-20-01117-t001:** Structure parameters. TMR—tunnel magneto-resistance.

Parameter	Value	Parameter	Value
Micro-cantilever (length × width × height (μm))	6000 × 1500 × 15	Density of micro-cantilever (g/cm^3^)	2.33
Permanent magnet (length × width × height (μm))	1500 × 1500 × 500	Density of magnet (g/cm^3^)	8.4
Coordinates of sensitive point ((x, y, z) (μm))	(1169, -198.5, 0)	Remanence (T)	1.023
Sensitivity of mechanical displacement (μm/g)	75.7	Resonant frequency (Hz)	62.37
Sensitivity of *Hx* (Oe/μm)	2.72	Linear region of *Hx* (μm)	±161
Sensitivity of *TMR* (mV/V/Oe)	0.31	Mechanical–thermal noise (ng/√Hz)	4.85

**Table 2 sensors-20-01117-t002:** The measurement results of the micro-system.

Parameter	Value	Parameter	Value
Measurement range (g)	±2	Sensitivity (mV/g)	181
Noise floor of the signal processing circuit (μg/√Hz)	0.2	Resolution (μg/√Hz at 1 Hz)	19
Bias stability (μg)	266	Bias instability (μg)	38

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
