# Peer review of "High-Precision Acceleration Measurement System Based on Tunnel Magneto-Resistance Effect†"

_sensors, 2020, doi:10.3390/s20041117_

Round 1

Reviewer 1 Report

The manuscript by Gao et al is certainly an intersting read and the work is very comprehensive and of good importance to the community. Science-wise, I believe it is qualified to be published on Sensors.

However, a mechanical flaw has to be addressed before its publication. The manuscript shares a significant overlap with their existing paper on IEEE 2019. Considering that was a conference proceeding, this is not acceptable. But the authors should have at least cited their own work, and spent some paragraphs discussing what is new or improved. I can see a fair amount of new information is indeed present, but not so apparent until I really dig into their previous paper. The authors should make some efforts to distinguish this work from the former ones.

Reviewer 2 Report

The paper deals with the design of a precise accelerometer based on tunnel magneto-resistance effect. In the first part of the paper the Authors are introducing the working principle of the sensor. Then, the design and the realization of the sensors is described in the following section. The obtained sensor has been experimentally validate and characterized.

The topic of the paper fits the scope of the journal.

The state of the art has been properly investigated.

The structure of the paper is clear and the language is proper.

I warmly suggest the Authors to include in the paper a summarizing table, where to include the measurement parameters of the designed sensor, including the accuracy, the resolution, the dynamic, and more.

The Authors should compare the obtained results with accelerometers using state of the art technologies.

Reviewer 3 Report

Dear Authors, The paper entitled "High Precision Acceleration Measurement System Based on Tunnel Magneto-Resistance Effect". In this paper, a high-precision tunnel magneto-resistance acceleration measurement system is proposed and demonstrated. Treat of a scientific work of a "force-magnetic-electric" coupling structure that converts an input acceleration into a change in magnetic field around the TMR sensor is designed. In such a structure, a micro-cantilever is integrated with a magnetic field source on its tip. Under an acceleration, the mechanical displacement of the cantilever causes a change in the spatial magnetic field sensed by the TMR sensor. The paper is original and innovative. But there are a few minor issues: Figure .1 needs to be improved as the information of the Direction of sensitivity is not visible. Figure 4. do not have the proper identification (a) and (b). Figure 6. the information is not visible.

Round 2

Reviewer 1 Report

I am happy with the modifications and this manuscript is good to go for Sensors.